# Quantum Measurements and Delays in Scattering by Zero-Range Potentials

**DOI:** 10.3390/e26010075

**Published:** 2024-01-16

**Authors:** Xabier Gutiérrez, Marisa Pons, Dmitri Sokolovski

**Affiliations:** 1Departamento de Química-Física, Universidad del País Vasco, UPV/EHU, 48940 Leioa, Spain; xabier.gutierrez@ehu.eus; 2EHU Quantum Center, Universidad del País Vasco, UPV/EHU, 48940 Leioa, Spain; marisa.pons@ehu.eus; 3Departamento de Física Aplicada, Universidad del País Vasco, UPV/EHU, 48013 Bilbao, Spain; 4IKERBASQUE, Basque Foundation for Science, 48011 Bilbao, Spain

**Keywords:** quantum measurements, zero-range potential, Eisenbud–Wigner–Smith delay, Larmor clock

## Abstract

Eisenbud–Wigner–Smith delay and the Larmor time give different estimates for the duration of a quantum scattering event. The difference is most pronounced in the case where the de Broglie wavelength is large compared to the size of the scatterer. We use the methods of quantum measurement theory to analyse both approaches and to decide which one of them, if any, describes the duration a particle spends in the region that contains the scattering potential. The cases of transmission, reflection, and three-dimensional elastic scattering are discussed in some detail.

## 1. Introduction

It is only natural to expect a quantum scattering process, be it collision between two particles or tunnelling across a potential barrier, to be characterised by a particular duration. Discussion about what this duration should be, and how ought to be measured, continues to date [1]. Traditionally, there were two schools of thought. One approach, originally proposed by Eisenbud and Wigner [2] and later extended by Smith to multichannel scattering [3], relies on the propagation of wave packet states and leads to time parameters expressed as the energy derivative of the phase of a scattering amplitude. An alternative method, first proposed by Baz’ [4] and later developed in [5], employs a spin, precessing in a small magnetic field introduced in the region of interest. The Larmor times, obtained in this manner, involve variations of scattering amplitudes in response to a small constant potential added to the region [6,7]. The authors of [5] concluded that the approach in [3] is in general incorrect, since both methods often lead to similar yet not identical results [5].

It is reasonable to ask whether the Eisenbad–Wigner–Smith (EWS) approach is merely wrong, or if, perhaps, one is dealing with two different yet equally valid methods. Similar questions have been asked, e.g., in [8,9], and more recently in [10]. The reader may also be interested in [11,12,13,14].

The appearance of both Eisenbud–Wigner–Smith and Larmor times may be strange to anyone used to the averages obtained with the help of a probability distribution, since neither of the two parameters look like conventional averages. The problem is most easily understood in terms of quantum measurement theory. In both cases, the particle is pre- an post-selected in its initial and final (transmitted) states. In both cases, one evaluates, in the standard way, an average of a variable expected to contain information about the duration spent in the barrier, a spin’s component, or the final particle’s position. A connection with quantum measurements is established once one notes that the probabilities used for the averaging are given by a convolution of an amplitude distribution with a kind of “apparatus function”. In the Larmor case, the amplitude distribution refers to the duration of a Feynman path spent in the barrier region, and the apparatus function is determined by the initial state of the clock (see, e.g., [15]). The EWS case is less obvious, but similar. The EWS amplitude distribution describes the range of spatial shifts with which a particle with a known momentum may emerge from the barrier, and the apparatus function is the envelope of the initial wave packet state (see, e.g., [16]). At this point, one notes an important role played by the Uncertainty Principle (UP) [17]. Since tunnelling can be seen as a result of destructive interference between alternatives, the presence of the apparatus function destroys this interference and, with it, the studied transition. The only way to preserve the transition is to make the apparatus function very broad, but then the UP would forbid one to distinguish between the durations or the shifts involved [17]. This is, indeed, the case, since if the perturbation is minimised, the measured average is expressed in terms of the first moment of an *amplitude* (and not a *probability*) distribution (also known as the “weak value” [18,19,20,21,22]). In addition to being complex-valued, the distribution may change sign, and the “weak value” does not faithfully represents the the range of values available to the transition [15]. For example, EWS time, measured in this manner, can turn out to be anomalously short, even though the barrier provides only for delays, compared to free propagation [16].

In this paper, we appeal to quantum measurement theory in order to analyse the similarities and the differences between both methods for determining the “tunnelling time” [23]. We will also show that the same approach can be applied to reflected particles, as well as to the case of potential scattering, originally studied in [2]. As an illustration, we consider particles scattered by a zero-range potential [24], chosen for two main reasons. Firstly, the disagreement between the Larmor and the Eisenbud–Wigner–Smith approaches is most pronounced in the ultra-quantum case where the particle’s de Broglie wavelength exceeds the size of the scatterer. Secondly, in both cases, the amplitude distributions on which our analysis is based have a particularly simple form, and the narrative can be abbreviated accordingly.

The rest of the paper is organised as follows. In Section 2, we describe two methods for measuring the duration τ a classical particle spends in a region containing the scattering potential. In Section 3, we briefly review the Larmor clock method and show that it predicts a zero delay whenever the size of the scatterer vanishes. In Section 4, we show that following the centre of mass of the scattered state leads to a different kind of “quantum measurement”. An inaccurate measurement of this kind determines an Eisenbud–Wigner–Smith time delay, which does not vanish for a zero-range potential. Section 5 and Section 6 analyse the centre-of-mass delay in transmission across a zero-range barrier or well. In Section 7, we extend the analysis to reflected particles. In Section 8, we consider elastic scattering by a zero-range spherically symmetric potential. Section 9 contains our conclusions.

## 2. Two Ways to Measure a Classical Duration

In classical mechanics, a particle always moves along a trajectory $x^{cl}\left(t\right)$, and the amount of time τ it spends in a region [a,b], containing a potential barrier (or a well) V(x) (see Figure 1), is a well-defined and useful concept. One way to measure it is to couple the particle to a clock, which runs only while the particle is in the region. This can be achieved by equipping the particle with a magnetic moment, introducing a magnetic field in [a,b], and dividing the angle of the precession ϕ by the Larmor frequency $\omega_{L}$. A simpler version of the Larmor clock is a pointer with position *f* and momentum λ, coupled to the particle while it is in the region. The full Hamiltonian of the system, therefore, is
(1)$$\begin{array}{c} H\left(x,p,f,\lambda \right)=p^{2}/2m+V\left(x\right)+\lambda \Theta_{\left[a,b\right]}\left(x\right), \end{array}$$
where *x* and *p* are the particle’s position and momentum, respectively, and $\Theta_{\left[a,b\right]}\left(x\right)$ is unity if *x* lies inside the interval [a,b] and 0 otherwise. Solving the Hamilton equations for λ=0, one easily finds the final pointer reading *f* equal to the duration sought:(2)$$\begin{array}{c} f\left(t\right)-f\left(0\right)=\int_{0}^{t}\Theta_{\left[a,b\right]}\left(x^{cl}\left(t^{\prime }\right)\right)dt^{\prime }=\int_{a}^{b}m^{1/2}dx/\sqrt{2[E-V(x)]}\equiv \tau \left(E\right), \end{array}$$
where E>V(x) is the particle’s energy.

The same quantity can be determined without the help of a clock, simply by comparing the current positions of the particle moving with and without the potential and defining (we reserve the subscript 0 for free motion)
(3)$$\begin{array}{c} \delta x^{cl}=x^{cl}\left(t\right)-x_{0}^{cl}\left(t\right) \end{array}$$
where $x_{0}^{cl}\left(t\right)$ is the trajectory with V(x)=0 (see Figure 2). Indeed, if both particles are launched simultaneously with equal initial momenta $p=\sqrt{2mE}$ from the same initial position, $x_{I}<a$, we have $x_{0}^{cl}\left(t\right)=vt+x_{I}$, where v=p/m. Both particles cross the region [a,b] since E>V(x). By the time the particle slowed down by a barrier reaches x=b, its faster free-moving counterpart will lie ahead by $|\delta x^{cl}|=v\left(\tau -\tau_{0}\right)$, where $\tau_{0}=\left(b-a\right)/v$ is the duration the free particle spends in the region. The difference in positions can, therefore, be used to evaluate the τ defined earlier in Equation (Equation 2):(4)$$\begin{array}{c} \tau \left(E\right)=\tau_{0}\left(E\right)-\delta x^{cl}/v \end{array}$$

A particle crossing a potential well, V(x)<0, spends in [a,b] a shorter duration, $\delta x^{cl}>0$, and $\tau \left(E\right)<\tau_{0}\left(E\right)$. Note that since both particles move freely for x>b, the distance between them remains the same once the region containing the potential is crossed. Note that this is also a kind of measurement, where the role of the pointer is now played by the particle itself. Both approaches can be generalised to quantum scattering, albeit with different results, and we will consider them separately.

## 3. The Quantum Larmor Time

The quantum analogue of Equation (Equation 2) has often been discussed (see, e.g., [25]), and we will describe it here only briefly. For a quantum particle, the transition amplitude between the states |ψ〉 and |ϕ〉 is given by the Feynman path integral (we use ℏ=1) $A\left(\phi ,\psi ,t\right)=\sum_{paths}\phi^{*}\left(x^{\prime }\right)\exp\left[iS\left(x,x^{\prime },t\right)\right]\psi \left(x\right)$, where *S* is the classical action functional, and $\sum_{paths}=\int dx^{\prime }\int dx\int_{\left(x,0\right)}^{\left(x^{\prime },t\right)}Dx\left(t\right)$ includes summation over all virtual paths connecting (x,0) with $\left(x^{\prime },t\right)$, as well as integration over the initial and final position. The classical expression in Equation (Equation 2) can be promoted to a functional on the Feynman paths, $t_{\left[a,b\right]}\left[x\left(t\right)\right]=\int_{0}^{t}\Theta_{\left[a,b\right]}\left(x\left(t^{\prime }\right)\right)dt^{\prime }$, and
(5)$$\begin{array}{c} A\left(\phi ,\psi ,t|\tau \right)=\sum_{paths}\delta \left(t_{\left[a,b\right]}\left[x\left(t\right)\right]-\tau \right)\times \phi^{*}\left(x^{\prime }\right)\exp\left[iS\left(x,x^{\prime },t\right)\psi \left(x\right)\right] \end{array}$$
yields the probability amplitude to complete the transition, while spending τ seconds in the chosen region [a,b]. Expressing the Dirac delta $\delta (t_{\left[a,b\right]}\left[x\left(t\right)\right]-\tau )$ as a Fourier integral allows one to rewrite (Equation 5) as
(6)$$\begin{array}{c} A\left(\phi ,\psi ,t|\tau \right)={\left(2\pi \right)}^{-1}\int_{-\infty }^{\infty }d\lambda \exp\left(i\lambda \tau \right)A_{V\left(x\right)+\lambda \Theta_{\left[a,b\right]}\left(x\right)}\left(\phi ,\psi ,t\right), \end{array}$$
where $A_{V\left(x\right)+\lambda \Theta_{\left[a,b\right]}\left(x\right)}\left(\phi ,\psi ,t\right)$ is the same transition amplitude, but in the modified potential $V\left(x\right)+\lambda \Theta_{\left[a,b\right]}\left(x\right)$. Note that there is a kind of uncertainty relation: to know τ, one needs to make the potential in [a,b] uncertain [15]. In the case of transmission (T) across the barrier (cf. Figure 1), the transition is between the same positive momentum states, |ψ〉=|ϕ〉=|p〉, over a long time, t→∞, and A(ϕ,ψ,t) is just the barrier’s transmission amplitude, T(p,V). Thus, quantum transmission is characterised by a range of durations, each endowed with an amplitude
(7)$$\begin{array}{c} A_{T}\left(p,\tau \right)={\left(2\pi \right)}^{-1}\int_{-\infty }^{\infty }d\lambda \exp\left(i\lambda \tau \right)T\left(p,V+\lambda \Theta_{\left[a,b\right]}\right). \end{array}$$

In its quantum version, the clock (Equation 1) becomes a von-Neumann pointer [26], coupled to the particle throughout its evolution. It may be prepared in an initial state |G〉, e.g., a Gaussian $G\left(f\right)=\langle f|G\rangle =C\exp\left(-f^{2}/\Delta f^{2}\right)$, centred at f=0. The pointer would be displaced by τ, G(f)→G(f−τ), if the value of τ were unique. With many such values, the final state of the clock is given by a sum over all displacements,
(8)$$\begin{array}{c} \Phi \left(f\right)=\int_{0}^{\infty }G\left(f-\tau \right)A_{T}\left(p,\tau \right)d\tau . \end{array}$$

Equation (Equation 8) defines the measurement briefly discussed in the Introduction.Equation (Equation 2) defines τ as the *net duration* spent by the particle in the barrier region.Since $T\left(p,V\right)=\int A_{T}\left(p,\tau \right)d\tau$, different durations interfere and cannot be told apart without a clock.The amplitude of finding the particle in a final state |p〉 and the clock’s pointer in |f〉 is the same as that of finding the particle in |p〉 provided the durations spent in the barrier are restricted to a range f−Δf≲τ≲f+Δf. One can say that τ has been measured to an accuracy Δf.

The clock is more accurate the smaller Δf is. It is also more perturbing, and sending Δf→0 would quench transmission, causing the particle to be reflected. In the opposite limit Δf→∞, an individual clock reading *f* provides little information. However, using (Equation 8), one can calculate the mean pointer reading (see Appendix A)
(9)〈f〉≡∫f|Φ(f)|2df∫|Φ(f)|2df→Δf→∞Re[τ¯[a,b](p)].
where ${\bar{\tau }}_{\left[a,b\right]}\left(p\right)$ is the “complex time” of [27],
(10)$$\begin{array}{c} {\bar{\tau }}_{\left[a,b\right]}\left(p\right)=\frac{\int_{0}^{\infty }\tau A_{T}\left(p,\tau \right)d\tau }{\int_{0}^{\infty }A_{T}\left(p,\tau \right)d\tau }=-i\partial_{\lambda }\left[\ln T(p,V+\lambda \Theta_{\left[a,b\right]})\right]{|}_{\lambda =0}. \end{array}$$

Evaluated with an alternating complex-valued distribution, the “weak value” [19] ${\bar{\tau }}_{\left[a,b\right]}\left(p\right)$ does not have the properties of a physical time interval, in agreement with the Uncertainty Principle [17], which forbids knowing the duration τ in precisely the same sense it forbids knowing the slit chosen by the particle in a double-slit experiment [15].

Our main interest here is in the delay (if any) experienced by a particle scattered by a zero-range potential,
(11)$$\begin{array}{c} V\left(x\right)=U\Theta_{\left[a,b\right]}\left(x\right),\;\left(b-a\right)\to 0,\\ \;U\to \infty ,\;U(b-a)=\Omega =const. \end{array}$$

As the region becomes ever more narrow, a→b, $t_{\left[a,b\right]}\left[x\left(t\right)\right]$ can only tend to zero for any smooth path x(t). However, Feynman paths are notoriously irregular [28], and a more rigorous justification will be given next. If the amplitude distribution for a free particle, V(x)=0, $A_{0}\left(p,\tau \right)$ is known, the distribution for a rectangular potential $V\left(x\right)=U\Theta_{\left[a,b\right]}\left(x\right)$, barrier or well, takes a particularly simple form:(12)$$\begin{array}{c} A_{T}\left(p,\tau \right)=\exp\left(-iU\tau \right)A_{0}\left(p,\tau \right). \end{array}$$

$A_{0}\left(p,\tau \right)$ can be computed by closing the integration contour in Equation (Equation 7) (with V=0) in the upper half of the complex λ-plane, where $T(p,\lambda \Theta_{\left[a,b\right]})$ has poles [6]. Only one pole contribution survives in the limit (b−a)→0, and using (Equation 12), one finds [6]
(13)$$\begin{array}{c} A_{T}\left(p,\tau \right)\underset{a\to b}{\to }\tau_{0}^{-1}\exp\left[-i\Omega \tau /\left(b-a\right)\right]\times \exp\left(-\tau /\tau_{0}\right), \end{array}$$
where $\tau_{0}=m\left(b-a\right)/p=\left(b-a\right)/v$. Since $|A_{T}\left(p,\tau \right)|$ tends to δ(τ), a measurement by a Larmor clock will always yield a zero duration for a very narrow potential (cf. Equations (Equation 7)–(Equation 10)). Next, we ask whether the same is true if one tries to deduce the same duration from the final position of a transmitted particle.

## 4. The Eisenbud–Wigner–Smith (Phase) Time

To obtain a quantum analogue of the procedure described by Equation (Equation 4), one can replace classical particles by wave packets (WPs) and evaluate the distance between their centres of mass (COM), as shown in Figure 3.

At t=0, both particles are prepared in the same state with a mean momentum *p*, a width Δx, and the COM at some $x_{I}<0$, $|x_{I}|>>\Delta x$. After scattering, the transmitted and the freely propagating state both lie to the right of the barrier. They are given by
(14)$$\begin{array}{c} \psi_{T}\left(x,t\right)=\int T\left(k,V\right)A\left(k,p\right)\exp\left(ikx-iE_{k}t\right)dk,\;E_{k}=k^{2}/2m, \end{array}$$
and
(15)$$\begin{array}{c} \psi_{0}\left(x,t\right)=\int A\left(k,p\right)\exp\left(ikx-iE_{k}t\right)dk=\exp\left(ipx-iE_{p}t\right)G_{0}\left(x,t\right), \end{array}$$
where $G_{0}\left(x,t\right)$ is an envelope of width Δx, initially peaking around $x=x_{I}$. For the separation between their COM, we have
(16)$$\begin{array}{c} \delta x_{com}^{T}\equiv \langle x\left(t\right)\rangle_{T}-\langle x\left(t\right)\rangle_{0},\;\langle x\left(t\right)\rangle_{T,0}\equiv \frac{\int x|\psi_{T,0}{|}^{2}dx}{\int |\psi_{T,0}{|}^{2}dx}, \end{array}$$
where, by Ehrenfest’s theorem [29],
(17)$$\begin{array}{c} \langle x\left(t\right)\rangle_{0}=pt/m+x_{I}\equiv vt+x_{I}. \end{array}$$

Throughout the paper we will consider what the authors of [8] called “completed events”, i.e., situations where both wave packets move freely to the right of the barrier, and the integration limits in Equation (Equation 16) can be extended to ±∞. There is no simple way to use the Feynman path integral, as was achieved in Equation (Equation 5). However, expressions similar to Equations (Equation 6)–(Equation 10) are readily obtained by rewriting Equation (Equation 14) as a convolution,
(18)$$\begin{array}{c} \psi_{T}\left(x,t\right)=e^{ipx-iE_{p}t}\int_{-\infty }^{\infty }G_{0}\left(x-x^{\prime },t\right)\eta_{T}\left(x^{\prime },p\right)dx^{\prime }, \end{array}$$
(19)$$\begin{array}{c} \eta_{T}\left(x^{\prime },p\right)=\frac{e^{-ipx^{\prime }}}{2\pi }\int_{-\infty }^{\infty }T\left(k,V\right)e^{ikx^{\prime }}dk, \end{array}$$
which conveniently separates the information of free motion, contained in $G_{0}$, from the properties of the scattering potential that determine $\eta_{T}\left(x^{\prime },p\right)$. If the spreading of the wave packet can be neglected, $G_{0}\left(x-x^{\prime },t\right)\approx G_{0}\left(x-x^{\prime },t=0\right)$, and Equation (Equation 4) defines a measurement different from the one described by Equation (Equation 8).

Note that $\exp\left(ipx\right)\eta_{T}\left(x^{\prime },p\right)∼\exp\left[ip\left(x-x^{\prime }\right)\right]$ allows one to *define*$x^{\prime }$ as the spatial shift with which a particle with momentum *p* emerges from the barrier (whatever this might mean).Since $T\left(p,V\right)=\int \eta_{T}\left(x^{\prime },p\right)dx^{\prime }$, different shifts interfere and cannot be told apart *a priori*.However, the amplitude of finding a particle, prepared at t=0 in a wave packet state (Equation 15), at a location *x* is the same as that of finding a particle, prepared in a state |p〉, in the same state |p〉, provided the shifts imposed by the barrier are restricted to a range $x-\Delta x≲x^{\prime }≲x+\Delta x$. Replacing the plane wave |p〉 with a wave packet (Equation 15) of width Δx and mean momentum *p* allows one to measure $x^{\prime }$ to an accuracy of Δx.

If the measurement is accurate, with Δx→0, one recovers the result for the free motion, $\psi_{T}\left(x,t\right)\approx \psi_{0}\left(x,t\right)$, since the high momenta that dominate the transmission are unaffected by the presence of the scattering potential. Tunnelling, as one may expect, is thereby destroyed. In the classical limit ℏ→0, E(p)>V(x), the rapidly oscillating $\eta_{T}\left(x^{\prime },p\right)$ develops a stationary region around $x^{\prime }=\delta x^{cl}=x^{cl}\left(t\right)-x_{0}^{cl}\left(t\right)$ (cf. Equation (Equation 3)), and the classical result is recovered [30].

The benefits of converting a transmission problem into a quantum measurement problem are most evident when discussing the properties of the so-called phase time (see, e.g., [31]). If $G_{0}\left(x,0\right)$ is very broad (the spreading can be neglected), interference between different shifts is not destroyed, and the Uncertainty Principle allows one to determine only a “complex shift”, $\bar{x^{\prime }}$, similar to the “complex time” in Equation (Equation 10). Like ${\bar{\tau }}_{\left[a,b\right]}\left(p\right)$ in Equation (Equation 10), it is obtained by averaging $x^{\prime }$ with a complex-valued distribution whose real and imaginary parts can change sign (Equation 19):(20)$$\begin{array}{c} {\bar{x^{\prime }}}_{T}\left(p\right)=\frac{\int_{-\infty }^{\infty }x^{\prime }\eta_{T}\left(x^{\prime },p\right)dx^{\prime }}{\int_{-\infty }^{\infty }\eta_{T}\left(x^{\prime },p\right)dx^{\prime }}=i\partial_{p}\left[\ln T(p,V)\right]; \end{array}$$
see also, e.g., [32,33,34]. Using Equation (Equation 16), one finds (see Appendix A)
(21)δxcomT→Δx→∞Rex′¯T(p).
where Rex′¯T(p) does not even have to lie in the region where $\eta_{T}\left(x^{\prime },p\right)\neq 0$. One may be tempted to convert the spatial delays into temporal ones using $\delta \tau =-x^{\prime }/v$. This is justified in a classically allowed case (cf. Equation (Equation 3)) but is unwarranted in general. Replacing δx in the classical Equation (Equation 4) by its quantum analogue (Equation 16) yields a “phase time” estimate for the duration spent in the barrier region,
(22)τphase(p)=(b−a)/v−Rex′¯T(p)/v=b−av+1v∂φT(p,V)∂p,
where φ(p,V) is the phase of the transition amplitude, $T\left(p,V\right)=|T\left(p,V\right)|\exp[i\varphi_{T}\left(p,V\right)]$, and $v^{-1}\partial \varphi_{T}\left(p,V\right)/\partial p=\partial \varphi_{T}\left(p,V\right)/\partial E$ is the Eisenbud–Wigner–Smith time delay.

The phase time (Equation 22), related to the “weak value” of the spacial shift $x^{\prime }$ in Equation (Equation 20) has the same problem as its Larmor counterpart (Equation 9). It can be anomalously short in tunnelling, even though the barrier only delays the particle relative to free propagation [30]. It does not grow as expected if the barrier width is increased [35].

It is, however, different in one important aspect. While the Larmor time vanishes for a zero-range potential, $\tau^{phase}\left(p\right)$ remains finite even as (b−a)→0. Next, we study how and why this happens.

## 5. Gaussian Wave Packets in a Zero-Range Potential

The transmission amplitude for a zero-range potential V(x)=Ωδ(x) (Equation 11) is well known to be
(23)$$\begin{array}{c} T\left(k,\Omega \right)=1-\frac{im\Omega }{k+im\Omega }. \end{array}$$

For Ω<0, its single pole in the complex plane of the momentum lies on the positive imaginary *k*-axis, where it corresponds to a bound state. For Ω>0, the pole moves to the negative *k*-axis, and closing in Equation (Equation 19) the contour of integration in the upper or lower half-plane, we obtain
(24)$$\begin{array}{c} \eta_{T}\left(x^{\prime },p\right)=\left\{\begin{array}{c} \delta \left(x^{\prime }\right)-\Theta \left(-x^{\prime }\right)m|\Omega |\exp(-ipx^{\prime }+m|\Omega |x^{\prime }),\mathrm{if}\;\Omega >0\\ \delta \left(x^{\prime }\right)-\Theta \left(x^{\prime }\right)m|\Omega |\exp(-ipx^{\prime }-m|\Omega |x^{\prime }),\mathrm{if}\;\Omega <0, \end{array}\right. \end{array}$$
where Θ(x)≡1 for x>0 and 0 otherwise.

Equations (Equation 24) provide a useful insight into how a potential acts on the incident particle. An incoming plane wave is multiplied by the transmission amplitude T(p,Ω) and, for a barrier Ω>0, we have
(25)$$\begin{array}{c} \exp\left(ipx\right)\to T\left(p,\Omega \right)\exp\left(ipx\right)=\exp\left(ipx\right)-m\Omega \int_{-\infty }^{0}\exp\left(m\Omega x^{\prime }\right)\exp\left[ip\left(x-x^{\prime }\right)\right]dx^{\prime }. \end{array}$$

Instead of providing a *temporal* delay for the transmitted particle (it could achieve this, e.g., by changing $\exp(ipx-iE_{p}t)$ into $\exp[ipx-iE_{p}\left(t-\tau_{p}\right)]$), a barrier acts as an “interferometer”, which splits the incoming plane wave into components with different phase shifts, corresponding to possible *spatial* delays, $x^{\prime }<0$. These delays are still present when the width of the barrier becomes zero, provided the strength of the potential increases accordingly (cf. Equation (Equation 11)). For a well, a similar expression contains additional plane waves, spatially *advanced* relative to free propagation. One classical feature survives in this purely quantum case. In some sense, a barrier (no bound states) tends to “delay” the particle, whereas a well tends to “speed it up”.

To quantify these effects, one can look at the motion of wave packets. As always, it is convenient to consider Gaussian states with a mean momentum *p* and a coordinate width Δx, (Δk=2/Δx):(26)$$\begin{array}{cc} & A\left(k,p\right)=2^{-1/4}\pi^{-3/4}\Delta k^{-1/2}\exp\left[-{\left(k-p\right)}^{2}/\Delta k^{2}-i\left(k-p\right)x_{I}\right],\\ & G_{0}\left(x,t\right)={\left[2\Delta x^{2}/\pi \sigma_{t}^{4}\right]}^{1/4}\exp\left[-{\left(x-vt-x_{I}\right)}^{2}/\sigma_{t}^{2}\right],\;\sigma_{t}\equiv {\left(\Delta x^{2}+2it/m\right)}^{1/2},\\ & |G_{0}\left(x,t\right)|={\left[2/\pi \Delta x_{t}^{2}\right]}^{1/4}\exp\left[-{\left(x-vt-x_{I}\right)}^{2}/\Delta x_{t}^{2}\right],\;\Delta x_{t}\equiv {\left(\Delta x^{2}+\Delta k^{2}t^{2}/m^{2}\right)}^{1/2}. \end{array}$$

The analysis is even simpler in the dispersionless case, $E_{k}=ck$, where the free amplitude undergoes no spreading:(27)$$\begin{array}{c} {\tilde{G}}_{0}\left(x,t\right)={\left[2/\pi \Delta x^{2}\right]}^{1/4}\exp\left[-{\left(x-ct-x_{I}\right)}^{2}/\Delta x^{2}\right]. \end{array}$$

The similarity between Equations (Equation 8) and (Equation 18) is yet more evident. The Larmor clock’s pointer state is displaced as a whole without spreading (cf. Equation (Equation 8)) because the kinetic energy, $\lambda^{2}/2\mu$, is omitted both in the classical Hamiltonian (Equation 1) and in its quantum counterpart (usually, by assuming the pointer’s mass μ to be large). A WP with no kinetic energy would not propagate at all, but making $E_{k}$ linear rather than quadratic in *k* has a similar effect.

## 6. Centre-of-Mass Delay for Transmission

Consider again Equation (Equation 16). According the Heisenberg’s Uncertainty Principle, a particle can have either a well-defined position or a well-defined momentum. Therefore, much depends on the coordinate width Δx of the initial Gaussian WP, as well as on the dispersion law.

The dispersionless case (Equation 18) is simpler, since the envelope (Equation 27), however narrow, is displaced without distortion (see Appendix B). As Δx→0, only the δ-term in Equation (Equation 24) needs to be taken into account, $\psi_{T}\left(x,t\right)\approx \psi_{0}\left(x,t\right)$, and $\delta x_{com}\to 0$. The contribution from the smooth part of $\eta_{T}$ vanishes, because $\int |{\tilde{G}}_{0}{\left(x\right)|}^{2}dx=1$ for any Δx, and $\int {\tilde{G}}_{0}\left(x\right)dx∼{\left(2\pi \Delta x^{2}\right)}^{1/4}\to 0$. A similar situation occurs in quantum measurements, where a singular δ-term in an amplitude distribution is responsible for the Zeno effect [36]. This is an expected result, since for |k|→∞, T(k,V)→1, and the potential has no effect on most of the momenta contained in the initial wave packet. In the opposite limit, Δx→∞, Δk→0, the singular term in Equation (Equation 24) can be neglected, and the “complex shift” in (Equation 20) is determined only by the smooth part of $\eta_{T}\left(x^{\prime },p\right)$,
(28)x′¯T(p)=−mΩp2+m2Ω2+im2Ω2p(p2+m2Ω2)≡Rex′¯T+iImx′¯T.

Its real part, $-m\Omega /\left(p^{2}+m^{2}\Omega^{2}\right)$, yields the distance between the COMs of the two WPs. The imaginary part of ${\bar{x^{\prime }}}_{T}\left(p\right)$ is related to the so-called “momentum filtering” effect, whereby the mean momentum of the transmitted WP is increased because higher momenta are transmitted more easily. Its significance is best illustrated in the case where dispersion is present, as we will discuss next.

For $E_{k}=k^{2}/2m$, we can consider scattering “completed” when the broadened transmitted WP lies sufficiently far to the right of the potential. The time needed for this can be estimated by following the motion of a free wave packet. We want the initial Gaussian WP to be placed as close as possible to the potential, e.g., at $x_{I}=-K\Delta x$, K>1. We also want to measure the COM as soon as the scattering is completed, i.e., when the COM of the freely propagating WP lies several widths away on the other side of the barrier, for example, at $x_{F}=K\Delta x_{t}$, where $\Delta x_{t}$ is defined in the last part of Equation (Equation 26). For a mean momentum *p*, the required time is easily found to be $t\left(p,\Delta k,K\right)=2mp\Delta xK/\left(p^{2}-K^{2}\Delta k^{2}\right)$. Note that we cannot simply send Δx→0, Δk→∞, as was possible for $E_{k}=ck$. In the limit Δx→∞, one finds (see Appendix C)
(29)δxcomT≈Rex′¯T(p)+Imx′¯T(k)Δk22mt(p,Δk,K)

The last fraction is clearly the excess mean velocity obtained through momentum filtering, absent for $E_{k}=ck$, where all momenta propagate with the same velocity *c*. The last term in Equation (Equation 29) behaves as ∼Δk∼1/Δx and was omitted in Equation (Equation 21). It may need to be retained for not-too-broad WPs, as is shown in Figure 4 for K=3.

Finally, we find the COM of the transmitted WP delayed by a zero-range barrier (δxT≈Rex′¯T<0 if Ω>0) and advanced by a zero-range well (δxT≈Rex′¯T>0 if Ω<0). This is not what happens in general, e.g., for a rectangular [16] or an Eckart barrier [30], where the COM of the greatly reduced transmitted WP is actually advanced. The reason for this is that in all such cases, T(k,V) has a large (infinite) number of poles in the complex *k*-plane, there are many exponential terms in the r.h.s. of Equation (Equation 24), and the resulting $\eta_{T}\left(x^{\prime },p\right)$ has a complicated form [30]. Although it vanishes for $x^{\prime }>0$, Gaussian envelopes in Equation (Equation 16) may interfere constructively in a small region of $x>x_{0}+vt$ and cancel each other elsewhere. This behaviour cannot, of course, be reproduced in the much simpler case studied here.

## 7. Centre-of-Mass Delay for Reflection

A similar analysis can be applied in the case of a particle, reflected (R) by a potential V(x), contained between x=a and x=b. The reflected WP is given by
(30)$$\begin{array}{c} \psi_{R}\left(x,t\right)=\int R\left(k,V\right)A\left(k,p\right)\exp\left(-ikx-iE_{k}t\right)dk, \end{array}$$
where R(k,V) is the reflection amplitude, satisfying ${\left|T\left(p,V\right)\right|}^{2}+{\left|R\left(p,V\right)\right|}^{2}=1$. It can also be written in a form similar to (Equation 18):(31)$$\begin{array}{c} \psi_{R}\left(x,t\right)=e^{-ipx-iE_{p}t}\int_{-\infty }^{\infty }G_{0}\left(-x-x^{\prime },t\right)\eta_{R}\left(x^{\prime },p\right)dx^{\prime }, \end{array}$$
where
(32)$$\begin{array}{c} \eta_{R}\left(x^{\prime },p\right)=\frac{e^{-ipx^{\prime }}}{2\pi }\int_{-\infty }^{\infty }R\left(k,V\right)e^{ikx^{\prime }}dk, \end{array}$$
and $G_{0}\left(-x-x^{\prime },t\right)$ (note x→−x) is the envelope of the mirror image of the free WP with respect to the origin, which is the same (except for a minus sign) as the envelope of a WP reflected by an infinite potential wall at x=0. One can still compare positions of the centres of mass, with and without the potential, by defining
(33)$$\begin{array}{c} \delta x_{com}^{R}=\langle x\left(t\right)\rangle_{R}+vt+x_{I},\langle x\left(t\right)\rangle_{R}\equiv \frac{\int x|\psi_{R}{|}^{2}dx}{\int |\psi_{R}{|}^{2}dx}. \end{array}$$

There is one complication not encountered in the case of transmission. Consider a potential $V_{s}\left(x\right)=V\left(x-s\right)$, obtained by displacing the original barrier or well by a distance *s*. Such a displacement has no effect on the transmission amplitude, $T\left(p,V_{s}\right)=T\left(p,V\right)$, but the reflection amplitude acquires an extra phase, $R\left(k,V_{s}\right)\to \exp\left(2iks\right)R\left(k,V_{s}\right)$, and $\eta_{R}\left(x^{\prime },p\right)$ changes into $\eta_{R}\left(x^{\prime }-2s,p\right)$. In other words, one needs to decide where to put the potential before making the comparison with free propagation. The ambiguity can be resolved by always placing the left edge of the potential at the origin, a=0 (see Figure 5), and considering the reflected particle *delayed* by the potential if $\delta x_{com}^{R}>0$, or *advanced* by it if $\delta x_{com}^{R}<0$.

With this agreed, a classical reflected particle can only be delayed, since it either bounces off the edge of the potential at x=0 or has to travel further to the right before making a U-turn. In the quantum case, this is not always true, as we will show next.

The reflection amplitude of a zero-range potential V(x,Ω)=Ωδ(x) is given by
(34)$$\begin{array}{c} R\left(k,\Omega \right)=\frac{-im\Omega }{k+im\Omega }, \end{array}$$
and
(35)$$\begin{array}{c} \eta_{R}\left(x^{\prime },p\right)=\left\{\begin{array}{c} -\Theta \left(-x^{\prime }\right)m|\Omega |\exp(-ipx^{\prime }+m|\Omega |x^{\prime }),\mathrm{if}\;\Omega >0\\ -\Theta \left(x^{\prime }\right)m|\Omega |\exp(-ipx^{\prime }-m|\Omega |x^{\prime }),\mathrm{if}\;\Omega <0. \end{array}\right. \end{array}$$

Note the absence of a δ-term, since R(k,Ω)→0 as |k|→∞. According to our convention, a reflected particle with a momentum *p* would be delayed by a zero-range barrier (as it would be in a classical case) and advanced by a zero-range well (a purely quantum effect, since there is no reflection from a well in the classical limit).

As was shown in the previous Section, without spreading (cf. Equation (Equation 27)), a narrow wave packet crosses a barrier or a well almost without reflection. Inserting (Equation 27) and (Equation 35) into Equation (Equation 31) for the small reflected part, we find (the upper sign is for a barrier) $|\psi_{R}{\left(x,t\right)|}^{2}∼\Theta \left(\pm x\mp ct\mp x_{I}\right)\Delta x\exp[-2m|\Omega |\left(\pm x\mp ct\mp x_{I}\right)]$, and
(36)$$\begin{array}{c} \delta x_{com}^{R}\underset{\Delta x\to 0}{\to }1/2m\Omega , \end{array}$$
as shown in Figure 6.

In the opposite limit of a broad WP we find
(37)δxcomR→Δx→∞Rex′¯R(p)=mΩp2+m2Ω2,
which is valid both with and without dispersion (cf. Equations (Equation 26) and (Equation 27)). In all cases, the reflected particle is delayed if Ω>0 and advanced if Ω<0.

## 8. Centre-of-Mass Delay in Elastic Scattering

Before concluding, we revisit the case of a particle scattered by a short-range spherically symmetric potential V(r) contained between r=0 and r=b. One can also think of a collision between two particles interacting via V(r), $r=|{\vec{r}}_{1}-{\vec{r}}_{2}|$. For a zero angular momentum, L=0, one can prepare a spatially symmetric wave packet ψ(r,t=0)$=\int A\left(k\right)\exp\left[-ik\left(r-r_{I}\right)\right]dk=\exp\left(-ipr\right)G_{0}\left(r,t=0\right)$, which converges on the scattering potential. The state ψ(r,t) satisfies a radial Schrödinger equation with a boundary condition ψ(r=0,t)=0, and one has a previously studied case of reflection, with an additional infinite wall added at r=0 (see Figure 7).

Proceeding as before, we write
(38)$$\begin{array}{c} \psi \left(r,t\right)=e^{ipr-iE_{p}t}\int_{-\infty }^{\infty }G_{0}\left(r-r^{\prime },t\right)\eta \left(r^{\prime },p\right)dr^{\prime },\\ \eta \left(r^{\prime },p\right)=\frac{e^{-ipr^{\prime }}}{2\pi }\int_{-\infty }^{\infty }S\left(k,V\right)e^{ikr^{\prime }}dk, \end{array}$$
where S(k,V), |S(k,V)|=1 is the scattering matrix element. For a zero-range potential, obtained in the limit $V\left(r\right)=U\Theta_{\left[0,b\right]}$, b→0, U→∞, $Ub^{2}\to const$ [5], S(k,V) is given by
(39)$$\begin{array}{c} S\left(k,\alpha \right)=-\frac{k+i\alpha^{-1}}{k-i\alpha^{-1}}=-\left[1+\frac{2i\alpha^{-1}}{k-i\alpha^{-1}}\right], \end{array}$$
where α is the scattering length [5], positive for a well, U<0, and negative for a barrier, U<0. The amplitude distribution $\eta (r^{\prime },p)$ becomes
(40)$$\begin{array}{c} \eta \left(r^{\prime },p\right)=\left\{\begin{array}{c} -\delta \left(r^{\prime }\right)+2\Theta \left(-r^{\prime }\right){\left|\alpha \right|}^{-1}\exp(-ipr^{\prime }+r^{\prime }/|\alpha |),\mathrm{if}\;\alpha <0\\ -\delta \left(r^{\prime }\right)+2\Theta \left(r^{\prime }\right){\left|\alpha \right|}^{-1}\exp(-ipr^{\prime }-r^{\prime }/|\alpha |),\mathrm{if}\;\alpha >0, \end{array}\right. \end{array}$$

In the limit of a broad nearly monochromatic WP, we find
(41)δrcom→Δx→0Rer′¯=2α1+k2α2
where the real-valued “complex shift”
(42)$$\begin{array}{c} \bar{r^{\prime }}=\frac{\int r^{\prime }\eta \left(r^{\prime },p\right)dr}{\int \eta (r^{\prime },p)dr}=-\frac{\partial \varphi (p,\alpha )}{\partial p} \end{array}$$
equals the Eisenbud–Wigner–Smith time delay, multiplied by the particle’s velocity *v*. There is no momentum filtering (cf. Equations (Equation 28) and (Equation 29)), since all momenta are perfectly reflected at the origin. As in the previous examples, a zero-range well (α>0) advances the scattered particle, while a zero-range barrier delays it.

## 9. Conclusions

We conclude with a brief summary of our results. Analysing a tunnelling (reflection, collision, etc.) time, one usually considers a quantum particle making a transition between known initial and final (transmitted, reflected, etc.) states and seeks to know what happened to the particle during the transition. To achieve this, one faces a familiar dilemma, best summarised in the double-slit example. In the absence of observation, both paths to a point on the screen interfere, and there is no telling which of the two slits was chosen by the particle [17]. An observation can determine the path but will destroy the interference pattern on the screen. The fact that one cannot know the path and observe the interference is referred to as the Uncertainty Principle [17].

There are many ways to decompose a transition amplitude, and in dealing with, say, a tunnelling event, one must make an educated choice. One option is to consider the duration the particle spends in the potential, and another is to analyse the distance between the transmitted and the freely propagated particle, once both have left the region that contains the potential. For a classical particle, both approaches give the same (expected) result, yet they disagree in the quantum case.

A decomposition in terms of the durations (τ) can be obtained by studying Feynman paths, as described in Section 3. A decomposition in terms of what is best described as “spatial delays” or “shifts” ($x^{\prime }$) experienced by the transmitted particle is also available (see Section 4). The problem is that both quantities are distributed with amplitudes, and the Uncertainty Principle forbids one to know which of the durations, or the shifts, actually occurred.

It is instructive to see how well-defined values are recovered in the classical limit. For a classically allowed transition, as ℏ→0, both amplitude distributions become highly oscillatory and develop narrow stationary points around the corresponding classical values [30]. Only the contributions from these regions survive in practice (e.g., by choosing a single shape *G* in Equation (Equation 8) or $G_{0}$ (Equation 15)), so one can speak of a unique duration the particle’s trajectory spends in a given region.

This is not the case in *tunnelling* across a wide, e.g., Eckart [30], barrier, where both amplitude distributions rapidly oscillate everywhere along their respective real axes. The transmission amplitude, which can be obtained by integrating the corresponding amplitude distribution over τ or $x^{\prime }$, is exponentially small owing to a precise cancellation. An attempt to restrict τ or $x^{\prime }$ to a particular range (not to mention, to a single value) would destroy the cancellation, and with it tunnelling. In this sense, tunnelling can be called a destructive interference phenomenon.

A further insight can be provided by quantum measurement theory. The authors of [5] are correct in saying that the Larmor clock measurements are related to the duration spent in the region (cf. Equation (Equation 5)). The clue is in the convolution formula (Equation 8. The likelihood of finding a clock’s reading *f* depends only on the amplitudes of having τ in the range f±Δf). What the authors of [5] appear to have failed to notice is that a *weakly perturbing* Larmor clock (broad *G* in Equation (Equation 8)) can only yield a “complex time” [27], essentially the first moment of an alternating amplitude distribution (cf. Equation (Equation 10)). The Uncertainty Principle warns against treating this quantity, or its parts, as physical time intervals [15].

The same problem occurs with the measurements relying on the transmitted particle’s position. A similar convolution formula (Equation 18) suggests that modulating a plane wave with a Gaussian envelope destroys the interference between the shifts experienced by the wave in the potential. The likelihood of finding the tunnelled particle at *x* depends only on the amplitudes of having $x^{\prime }$ in the range x±Δx. If one tries to preserve the interference by increasing either Δf or Δx, one is only able to measure a “complex spatial delay”, again the first moment of a different alternating amplitude distribution.

It should be noted that a distance separating the COM of the free and transmitted states (as seen on a snapshot) can always be converted into the time delay, Δt, with which the transmitted particles arrive, e.g., at a fixed detector. In general, it is unwise to relate the observed Δt to the excess duration (positive or negative) spent by the particle in the potential. The price is having to deal with “tunnelling times” so short as to defy Einstein’s causality, see, e.g., [37]. Such a relation exists only in the presence of a unique classical trajectory, as we have already discussed above. It cannot be invoked when the Uncertainty Principle conceals the details of the past of a tunnelling particle.

Zero-range potentials offer straightforward illustrations of both methods, due to the simplicity of the transmission amplitude (Equation 23). Here, the results of both measurements differ from those obtained for broad barriers or wells [30]. They also contradict each other directly. One expects a particle, classical or quantum, to be able to spend a vanishing duration in a region of a vanishing volume. This was confirmed by a direct calculation in Section 3, showing the amount of time spent in a zero-range potential to be precisely zero. There is no interference to destroy, and one can use a Larmor clock of any accuracy, Δf, to obtain the result τ=0 without perturbing the transition.

Something different occurs if one uses the centre of mass of a broad wave packet as a “pointer” in order to measure the shift experienced by the particle with a momentum *p* in the potential. Apart from the singular term $\delta (x^{\prime })$, the amplitude distribution in Equation (Equation 24) contains a single exponential tail, extending to the positive and negative *x* for a well (Ω<0) and a barrier (Ω>0), respectively. Thus, the COM of a broad wave packet will be either delayed or advanced relative to free propagation by Rex′¯T in Equation (Equation 28). However, one should not be tempted to “explain” this effect by claiming that the particle spends a shorter or a longer duration in a region where the only possible duration must be zero.

Experimental techniques for detecting the arrival of wave packets are currently available [38]. A recent realisation of a weak Larmor clock was reported in [39]. Both methods can, in principle, be applied in a situation where the particle’s de Broglie wavelength exceeds the size of the scatterer. An analysis in terms of quantum measurement theory allows one to identify two different measured quantities, as well as the conditions under which they are measured. It also helps to avoid a spurious contradiction between the zero duration registered by the Larmor clock and an apparently non-zero result seen in wave packet propagation.

Finally, we demonstrated that a similar analysis can be applied to the case of reflection, or to a particle scattered by a spherically symmetric potential. The latter case, we recall, was studied in the early papers on the subject [2,4]. Transmission, reflection, and collision times, best understood in light of the Uncertainty Principle, have a place in the conventional formalism of elementary quantum mechanics.

## Figures and Tables

**Figure 1 entropy-26-00075-f001:**
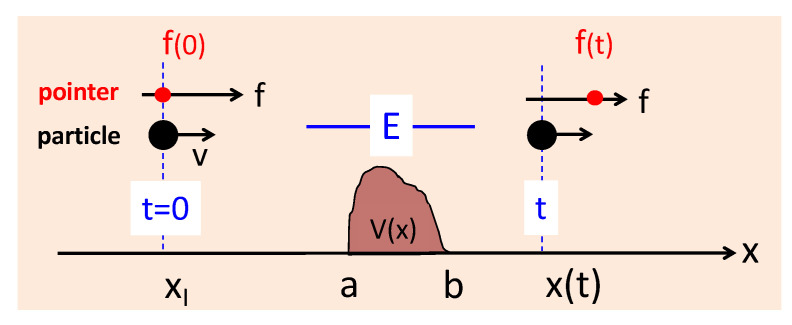
A classical particle is coupled to a clock (Equation (Equation 1)), which runs only while it is inside the potential. The duration the particle has spent in [a,b] can be read off the pointer’s position (cf. Equation (Equation 2)).

**Figure 2 entropy-26-00075-f002:**
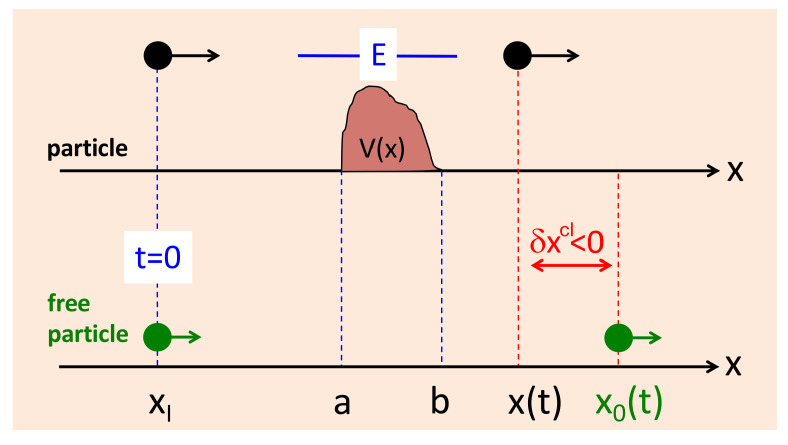
The same duration can be evaluated by launching two classical particles, one with and one without the potential, and comparing their positions once they have crossed the region [a,b] (cf. Equation (Equation 4)).

**Figure 3 entropy-26-00075-f003:**
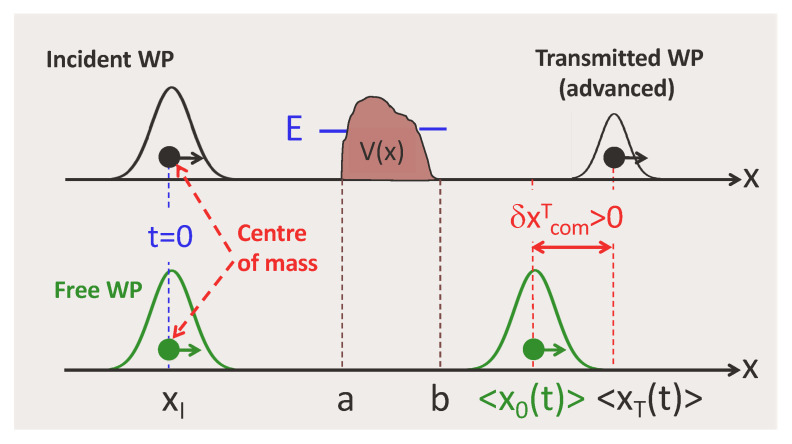
A quantum analogue of Figure 2: classical particles are replaced by wave packets, whose centres of mass are used as the reference points. The advancement, or delay, of the transmitted particle results from the interference between all virtual spatial shifts provided by the potential (cf. Equation (Equation 18)).

**Figure 4 entropy-26-00075-f004:**
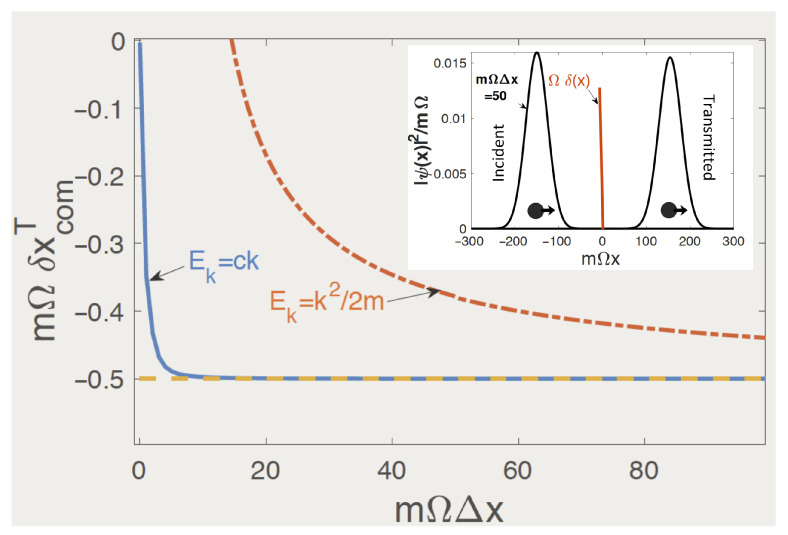
Centre-of-mass delay for transmission by a zero-range barrier, Ω>0, with and without dispersion vs. WP’s width Δx. Shown by the dashed line is the delay in the limit Δx→∞ (cf. Equation (Equation 29)). The inset shows the initial and final wave packets for mΩΔx=50 and $E_{k}=k^{2}/2m$. The parameters used are Ω/c=mc/p=1, $x_{I}/\Delta x=3$ and $m\Omega^{2}t\left(p,\Delta k,3\right)=6m\Omega \Delta x/\left[1-36{\left(m\Omega \Delta x\right)}^{2}\right]$.

**Figure 5 entropy-26-00075-f005:**
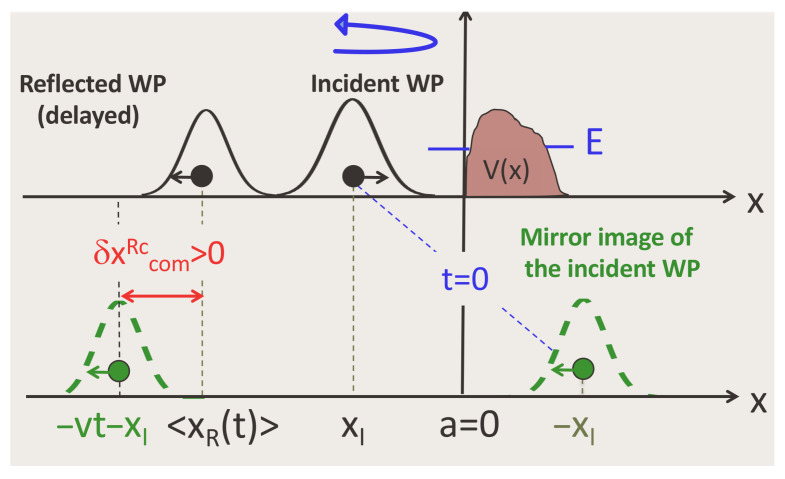
The position of a quantum particle, reflected by a potential V(x), is compared with that of a free particle launched in the opposite direction from $-x_{I}>0$. The particle is said to be *delayed* by the potential if its COM lies to the right of the COM of the freely propagating WP, or *advanced* if the opposite is true (cf. Equation (Equation 33)).

**Figure 6 entropy-26-00075-f006:**
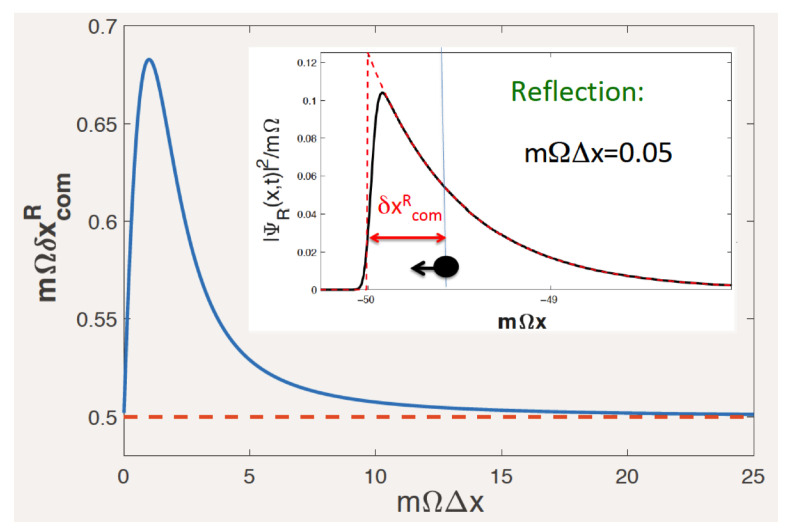
Centre-of-mass delay for reflection by a zero-range barrier, Ω>0, in the absence of dispersion, $E_{k}=ck$, (cf. Equation (Equation 27)) vs. WP’s width Δx. Shown by the dashed line is the delay in the limit Δx→∞ (cf. Equation (Equation 37)). The inset shows the small reflected WP (solid) and its limiting form for Δx→0 (dashed). The parameters used are Ω/c=mc/p=1, $m\Omega x_{I}=-50$, and mΩct=100.

**Figure 7 entropy-26-00075-f007:**
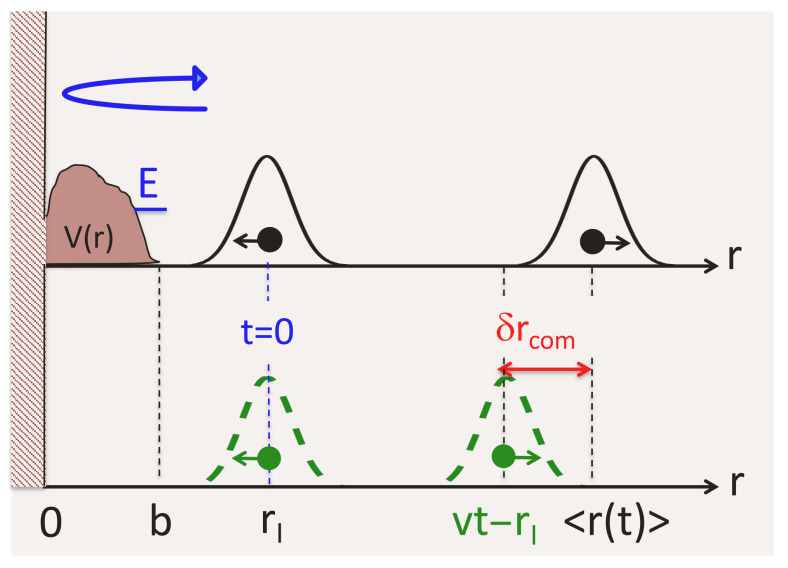
A quantum particle with zero angular momentum, L=0, is scattered by a spherically symmetric potential V(r). The boundary condition at the origin is equivalent to putting an infinite wall at r=0.

## Data Availability

No new data were created or analyzed in this study. Data sharing is not applicable to this article.

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
