# Peer review of "Quantum Measurements and Delays in Scattering by Zero-Range Potentials"

_entropy, 2024, doi:10.3390/e26010075_

Round 1
Reviewer 1 Report
Comments and Suggestions for Authors
The present paper compares two different approaches for evaluating the duration spent by a quantum particle inside a barrier, commonly known as "tunneling time." This comparison involves the Wigner time-delay method and the Larmor clock method, both widely employed in theoretical and experimental investigations, despite their potential to yield disparate results. Due to the ongoing controversy surrounding the concept of tunneling time, the authors prefer to examine the spatial shift of a wave packet resulting from its propagation within the barrier.
To gain a comprehensive understanding of the distinctions and similarities between these approaches, the authors concentrate on the delta-function barrier. In this scenario, the tunneling time according to the Larmor clock method is consistently zero, in contrast to the predictions of the Wigner time-delay method. Furthermore, this limiting case allows the authors to derive straightforward analytical results that prove beneficial for comprehending the measurement process.
The authors employ the Feynman path integral method, extended for evaluating the particle's duration within the potential, a methodology developed by one of the authors long ago (Ref. 21). Additionally, the authors correlate their findings with measured averages of the pointer, essential for accurately determining the spatial shift of wave packets and its connection to tunneling time.
I believe the authors' analysis is sound, providing a clear explanation of the differences between the two evaluation methods for interaction time. This contribution holds significance for enhancing comprehension in measurement theory and tunneling time, and I recommend it for publication with the suggestion that the authors consider my recommendations for improving the presentation.
On page 8 (lines 231-232), the authors state, "In some sense, a barrier tends to 'delay' the particle, where a well tends to 'speed it up." This point lacks clarity. Specifically, if the well contains a bound state, it can trap the particle inside, thereby delaying any potential speed-up in particle propagation. It would be beneficial if the authors could elaborate on the origin of this "speed up."
Another aspect requiring further clarification is related to the Uncertainty Principle, which plays a pivotal role in the authors' arguments. On page 2 (lines 40-42), the authors assert, "Since tunnelling can be seen as a result of destructive interference between alternatives, the presence of the apparatus function destroys this interference." This assertion is unclear. Interference, whether destructive or constructive, between different components of any wave packet typically governs its propagation. Why do the authors presume that inside the barrier, only destructive interference occurs in the wave packet? How does this differ from propagation outside the barrier? Even if the authors have touched on these points in previous works, it is crucial to elaborate more on these aspects in the current paper for greater clarity.
Reviewer 2 Report
Comments and Suggestions for Authors
Paper Summary
The paper titled "Quantum measurements and delays in scattering by zero-range potentials" by X. Gutiérrez, M. Pons, and D. Sokolovski discusses the different approaches to measuring the duration of a quantum scattering event. The authors compare the Eisenbud-Wigner-Smith (EWS) delay, which is based on wave packet propagation, with the Larmor time, which involves the precession of a spin in a magnetic field. They analyze both methods using quantum measurement theory and determine which one accurately describes the duration a particle spends in the region containing the scattering potential. The document focuses on the cases of transmission, reflection, and three-dimensional elastic scattering. The authors also consider particles scattered by a zero-range potential, which highlights the disagreement between the EWS and Larmor approaches. The paper is organized into sections that discuss the different measurement methods, analyze the center-of-mass delay in various scenarios, and provide conclusions.
Advantages:
1. The paper provides a detailed analysis of two methods for measuring the "tunnelling time" of particles in a barrier region.
2. It discusses the similarities and differences between the Larmor clock method and the Eisenbud-Wigner-Smith method.
3. The paper considers the case of particles scattered by a zero-range potential, which allows for a simplified analysis.
4. It explores the effects of potential barriers and wells on the motion of wave packets.
5. The paper presents mathematical equations and formulas to support its analysis.
Disadvantages:
1. The paper does not provide experimental results or empirical evidence to support its claims.
2. It is not clear how the findings of this research can be applied in practical applications or real-world scenarios.
3. Has the effectiveness of quantum measurements been considered in cases other than particles scattered by zero-range potentials?
4. Ensure that the referenced literature is up-to-date, especially in the fields of quantum scattering and quantum measurement theory.
5. Consider citing interdisciplinary research, particularly those that may offer additional insights or support your arguments.
Reviewer 3 Report
Comments and Suggestions for Authors
This paper discusses the question of evaluating reflection and transmission “durations“ due to scattering by Dirac-delta-type potentials. The results are correct, and the proposed interpretations are sound while aiming at clarifying controversies about the determination of tunneling times.
The main results are well-explained, which should render the conclusions accessible to a broad readership. However, at times, the authors skip several steps at once to reach them, which may make the calculations hard to follow for people who don’t already know in some detail how these computations generally work out for other types of potentials. Although not imperative, I would recommend that the authors consider putting a few more explanations in places where they provide valid statements without immediate proof.
I have a few minor comments that require the authors’ attention before publication.
- Lines 41 – 43 : The meaning of sentence “Since tunnelling can be seen as a result of destructive interference between alternatives, the presence of the apparatus function destroys this interference and, with it, the studied transition” is not clear. What do the authors mean by “tunnelling can be seen as a result of destructive interference”? (Surely, what was transmitted did not interfere destructively.) What do the authors mean by “the presence of the apparatus function destroys this interference”? (The apparatus function is always present in my understanding. Indeed, this seems contradictory with the second part of the argument, starting line 43, where the authors state that a broad apparatus function preserves the interferences.)
- Line 80: I recommend explaining the physical role played by λ in this measurement scheme to improve clarity (especially as it is set to 0 and does not appear in equation 2).
- Line 115: “Note that there is a kind of uncertainty relation: to know τ one needs to make the potential in [a, b] uncertain.” This should be explained and/or a reference should be provided.
- Line 187: Is it really “∆f → 0” in the quantum case for section 4; shouldn’t it be ∆x?
- Line 198: The authors should define what they mean by an alternating distribution.
- Line 216: The authors state that they set the mass to unity. This should be avoided as the mass is a physical parameter and not a physical constant. This is confusing, especially since the mass is preserved in many equations throughout the text (for example, in Fig 5 and 6 or equation 26). The authors must reintroduce the mass where they set it to 1 (this is equivalent to replacing Ω by Ω m in most equations involving Ω in sections 5, 6, and 7. (Eqs. 23, 24, 25, 28; Line 258; Eqs. 34, 35; Line 317; Eqs. 36, 37)
- Line 327: In section 8, the authors study the scattering in a spherically symmetric setting (spherical potential and spherically symmetric converging wave). This section would gain from a discussion of the physical relevance of such a setting, in particular the consideration of a converging 3D wave.
- Line 391: “if G(x) is replaced by G0-tilde”. Shouldn’t it be G0 (without the tilde)? In any case, if the reader goes to Annex A when it is first referenced G0-tilde is not yet defined.
- Equation A3: To deduce A3 from A1, it seems that the level of approximation should be different in the numerator and denominator (first order in G’ in the numerator, no G’ in the denominator). The author should explain why this is warranted.
- Appendix B requires a few corrections and clarifications. In particular, equation 8 is currently wrong (the lhs is a ket, while the rhs is a number; the projection in the rhs is realized on an undefined ket phi_0). Also, using Sum_k ıphi_R(k)><phi_R(k)ı+ ıphi_L(k)><phi_L(k)I as a resolution of the identity to define ıpsi_0> would require these states to be normalized (which does not seem to be the case in A6, A7).
I also noticed a few minor formatting problems:
- The references are not numbered by order of appearance (line 20, 150, 151)
- Eq. 4 and Fig. 2 use δx; shouldn’t it be δx^cl?
- Eq. 5: there is a missing bracket in the exponential of the action (should be Exp[i S(x,x’,t)])
Once these points are taken into account, I recommend publication considering the merits of the paper.
Comments on the Quality of English LanguageI found a few typos and orthographic or grammatical problems
- Line 38: “spacial” (spatial)
- Line 53: grammatical problem in “we use the measurement theory techniques”
- Line 55: grammatical problem in “as well as in the case of potential scattering”
- Line 59: “is is” (is)
- Line 76: “well defined” (well-defined)
- Line 109: “on the Feynman’s paths” (on the Feynman paths)
- Line 133: “The clock is the more accurate the smaller is ∆f” (The clock is more accurate the smaller ∆f is)
- Line 164: there are two commas after ∆x
- Line 174: there is presently a circle above the letter I in the “If” at the beginning of the sentence
- Lines 175-176: grammatical problem in “and Eq. (4) defines the measurement different from the one described by Eq.(8). ”
- Line 195: “If the G0(x,0)” (If G0(x,0))
- Line 212: “How an why” (How and why)
- Line 326: “specially” (spatially)
- Line 443: “elswhere” (elsewhere)
Round 2
Reviewer 1 Report
Comments and Suggestions for Authors
The authors accept all my recommendations in the revised manuscript.
I therefore recommend the publication in its present form. I only propose to replace the text "no bound state" on line 234 by " no resonance state". I think that it was the authors' intention.
Reviewer 2 Report
Comments and Suggestions for Authors
The authors have fully revised the manuscript, making it suitable for publication.
Reviewer 3 Report
Comments and Suggestions for Authors
The authors have addressed the issues, so I recommend publication.
There remains a small issue with reference numbering (line 20).